# Bagel: A Benchmark for Assessing Graph Neural Network Explanations

## Abstract

Evaluating interpretability approaches for graph neural networks (GNN) specifically is known to be challenging due to the lack of a commonly accepted benchmark. Given a GNN model, several interpretability approaches exist to explain GNN models with diverse (sometimes conflicting) evaluation methodologies. In this paper, we propose a benchmark for evaluating the explainability approaches for GNNs called BAGEL. In BAGEL, we firstly propose four diverse GNN explanation evaluation regimes – 1) *faithfulness*, 2) *sparsity*, 3) *correctness*, and 4) *plausibility*. We reconcile multiple evaluation metrics in the existing literature and cover diverse notions for a holistic evaluation. Our graph datasets range from citation networks, document graphs, to graphs from molecules and proteins. We conduct an extensive empirical study on four GNN models and nine post-hoc explanation approaches for node and graph classification tasks. We release both the benchmarks and reference implementations and make them available at https://anonymous.4open.science/r/Bagel-benchmark-F451/.

## 1 Introduction

Graph neural networks (GNNs) (Veličković et al., 2018; Kipf & Welling, 2017; Klicpera et al., 2019; Xu et al., 2019; Hamilton et al., 2017) are representation learning techniques that encode structured information into low dimensional space using a feature aggregation mechanism over the node neighborhoods. GNNs have shown state-of-the-art performance across many scientific fields in various important downstream applications, such as molecular data analysis, drug discovery, toxic molecule detection, and community clustering (Dong et al., 2022; Gaudelet et al., 2021; Ying et al., 2018).

There have been benchmarks and datasets for interpretability of machine learning models (Wiegreffe & Marasović, 2021; Liu et al., 2021). The rising number of applications of GNNs in several sensitive domains like medicine and healthcare (Dong et al., 2022; Lu & Uddin, 2021) necessitates the need to explain their decision-making process. GNNs are inherently black-box and non-interpretable. Moreover, due to the complex interplay of node features and neighborhood structure in the decision-making process, general explanation approaches (Lundberg & Lee, 2017; Ribeiro et al., 2016; Singh & Anand, 2020) cannot be trivially applied for graph models. Consequently, several explanation technique (Ying et al., 2019; Funke et al., 2022; Vu & Thai, 2020; Yuan et al., 2020; Schnake et al., 2021; Huang et al., 2020; Schlichtkrull et al., 2020; Yuan et al., 2021) have been proposed for GNNs in the last few years. A known challenge in developing explanation techniques is that of evaluation of the quality of explanations. This challenge also extends to the evaluation of explainability approaches for GNNs and is the focus of this work.

Existing approaches usually focus on a certain aspect of evaluation, sometimes even performed on synthetic datasets. For example, some works employ synthetic datasets with an already-known subgraph (sometimes referred to as the ground truth reason or simply the ground truth). Explanations are then evaluated based on their agreement with the ground truth. Such an evaluation is sometimes flawed as one cannot always guarantee that the GNN has used in the first place the seeded subgraph for its decision-making process Faber et al. (2021). Besides, there is no standardized procedure for comparing different GNN explanations. For example, feature attribution methods can generate soft masks (feature importance as a distribution) or hard masks (boolean selections) over features. Comparing hard and soft mask explanations needs a common and

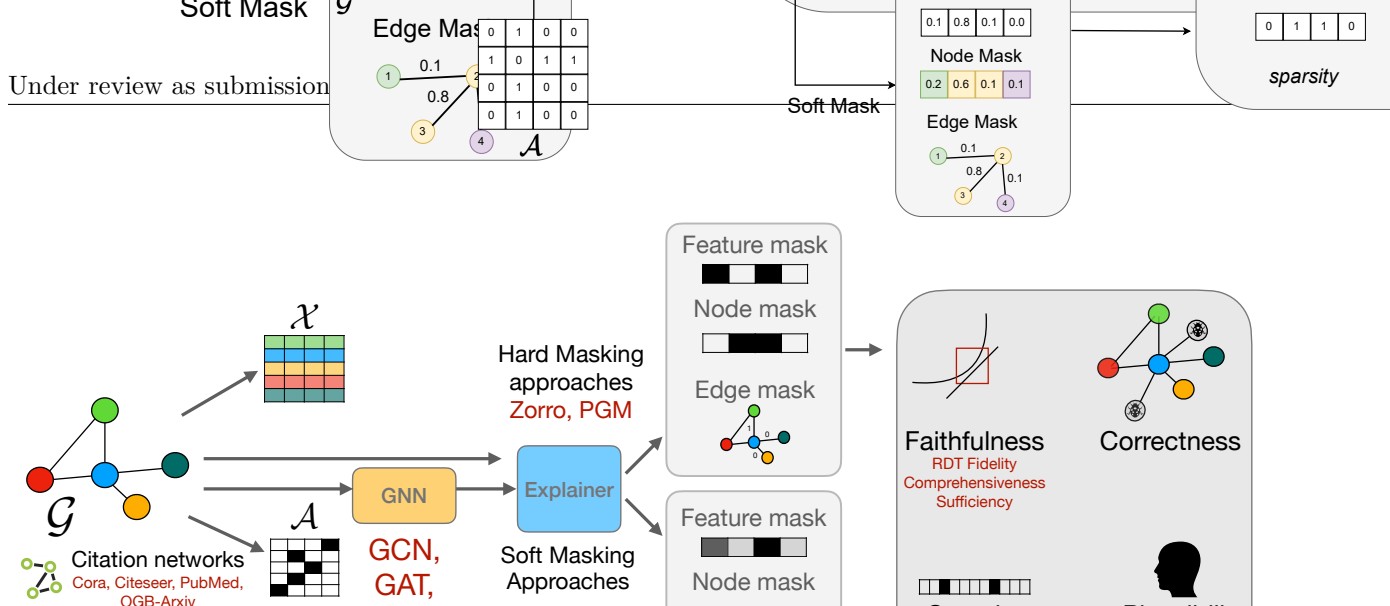

Figure 1: An overview of the BAGEL benchmark.

standardized protocol. Finally, the check for *human plausibility* and correctness have been ignored in the evaluation of GNN explainers. Human plausibility checks if a model predicts *right for the right reason.* On the other hand, the correctness of an explanation checks if the explainers is able to isolate spurious correlations and biases that are intentionally added to the training data as a proxy for biases present in real-world data.

To address the issues of a holistic evaluation and contribute a resource to the growing community on GNN explainability, we developed BAGEL, a benchmark platform for evaluating explanation approaches for graph neural networks or GNNs. BAGEL as depicted in Figure 1 covers diverse datasets (Sen et al., 2008; Debnath et al., 1991; Borgwardt et al., 2005; Zaidan & Eisner, 2008) from the literature, a range of standardized metrics, and a modular, extendable framework for execution and evaluation of GNN explanation approaches, along with initial implementations of recent popular explanation methods. In Table 1, we show the combination of metrics and datasets that we use in our benchmark. BAGEL includes:

○ Four diverse evaluation notions that evaluate the *faithfulness, sparsity, correctness, and plausibility* of GNN explanations on real-world datasets. While the first three metrics focus on evaluating the explainers, plausibility checks for explanations to be human congruent.

○ Besides the widely used datasets for measuring faithfulness of explanations, BAGEL consists of new datasets for the plausibility of explanation approaches in our benchmark datasets.

○ We unify multiple evaluations, metrics, domains, and datasets into an easy-to-use format that reduces the entry barrier for evaluating new approaches to explain GNNs. Additionally, we provide an extendable library to implement and evaluate GNN explainers.

○ We conduct an extensive evaluation of GNNExplainer(GNNExp) (Ying et al., 2019), PGM-Explainer(PGM) (Vu & Thai, 2020), Zorro (Funke et al., 2022), Grad (Simonyan et al., 2013), GradInput(Simonyan et al., 2013), Integrated Gradient(IG) (Sundararajan et al., 2017), SmoothGrad (Smilkov et al., 2017), CAM (Pope et al., 2019) and GradCAM (Pope et al., 2019) in BAGEL.

We show that there is no clear winner in GNN explanation methods showing nuanced interpretations of the GNN explanation methods using the multiple metrics considered. We finally note that evaluating the effectiveness of explanations is an intrinsically human-centric task that ideally requires human studies. However, the goal of BAGEL is to provide a fast and accurate evaluation strategy that is often desirable to develop new explainability techniques using empirical evaluation metrics be-

Table 1: Datasets and metrics.

| Task | Dataset | Metric | | | | |
| | | *Faithfulness* | | Sparsity | Correctness | Plausibility |
| | | RDT-Fidelity | Suff. & Comp. | | | |
|---|---|---|---|---|---|---|
| Node | CORA | ✓ | ✗ | ✓ | ✓ | ✗ |
| | CITESEER | ✓ | ✗ | ✓ | ✓ | ✗ |
| | PUBMED | ✓ | ✗ | ✓ | ✗ | ✗ |
| | OGBN-ARXIV | ✓ | ✗ | ✓ | ✗ | ✗ |
| | Synthetic | ✓ | ✗ | ✓ | ✓ | ✓ |
| Graph | Movie Reviews | ✗ | ✓ | ✗ | ✗ | ✓ |
| | MUTAG | ✓ | ✓ | ✓ | ✗ | ✗ |
| | PROTEINS | ✓ | ✓ | ✓ | ✗ | ✗ |
| | ENZYMES | ✓ | ✓ | ✓ | ✗ | ✗ |

fore the human trial stage. The code and the datasets used in our benchmark are available at
https://anonymous.4open.science/r/Bagel-benchmark-F451/.

## 2 Background and Preliminaries

**Graph Neural Networks.** Let $\mathcal{G}(\mathcal{V}, \mathcal{E})$ be a graph with $\mathcal{V}$ is a set of nodes and $\mathcal{E}$ is a set of edges. Let $\mathcal{A} \in \{0,1\}^{(n,n)}$ be the adjacency matrix of the graph where $n$ is the number of nodes in the graph with $\mathcal{A}_{ij} = 1$ if there is an edge between node $i$ and $j$ and 0 otherwise. Let $\mathcal{X} \in \mathbf{R}^{(n,d)}$ be the features matrix where $d$ is the number of features. For a given node $v \in \mathcal{V}$, $\mathbf{x}_v$ denotes its features vector and $\mathcal{N}_v$ is a set of its neighbors. We denote the trained GNN model as $f$ on the given graph. For each layer $\ell$, the representation of node $v$ is obtained by aggregating and transforming the representations of its neighboring nodes at layer $\ell - 1$

$$\boldsymbol{h}_v^{(\ell)} = \text{AGG}\left(\left\{\mathbf{x}_v^{(\ell-1)}, \left\{\mathbf{x}_u^{(\ell-1)} \mid u \in \mathcal{N}_v\right\}\right\}\right), \quad \mathbf{x}_v^{(\ell)} = \text{TRANSFORM}\left(\boldsymbol{h}_v^{(\ell)}, W^{(\ell)}\right), \tag{1}$$

where $W^{(\ell)}$ represents the weight matrix at layer $\ell$. The aggregation function AGG function is dependent on the GNN model. For example, graph convolution network (GCN) (Kipf & Welling, 2017) uses a degree weighted aggregation of neighborhood features, whereas graph attention network (GAT) (Veličković et al., 2018) learns neighborhood weights via an attention mechanism. The prediction can be obtained at the final layer using a *softmax function*. An additional pooling layer is applied for *graph classification* tasks before applying *softmax function*.

**Computational Graph.** We note that for the task of node classification, for any node $v$ the subgraph taking part in the computation of neighborhood aggregation operation, see (1), fully determines the information used by GNN during inference time to predict its class. In particular, , for a $k$-layer GNN, we refer to the subgraph induced on nodes in the $k$-hop neighborhood of $v$, as its *computational graph*. Note the for the task of graph classification the computational graph would be the entire graph.

### 2.1 Post-hoc explanations and evaluation for GNNs

**GNN Explanation.** Post-hoc explainers for GNNs produce feature and local structure attributions where a combination of a masked set of nodes, edges, and features is retrieved as an explanation. To compute the explanation for a $k$-layer GNN, the $k$-hop neighborhood (i.e.its computational graph) of the node is utilized. For an explanation $S$, the explanation mask $M(S)$ is computed over the input nodes/edges and the features in the computational graph. Note that $M(S)$ could be binary or a continuous mask and contains the importance scores for the corresponding nodes/features. We note that as different explainers either return node or edge importance scores, for consistent comparison we convert edge masks to node masks.

BAGEL currently consists of 3 classes of post-explanation techniques: *gradient based*, *perturbation based* and *surrogate model* approaches. The gradient-based methods (Simonyan et al., 2013; Sundarararajan et al.,

2017; Smilkov et al., 2017; Pope et al., 2019) are the simplest approaches for generating the explanation for any differentiable trained model. In these approaches, the importance scores for the explanation are usually computed using gradients of the input. The perturbation-based approaches (Funke et al., 2022; Ying et al., 2019; Luo et al., 2020; Yuan et al., 2021; Schlichtkrull et al., 2020) learns the important features and structural information by observing the predictive power of the model when noise is added to the input. The surrogate-based approaches (Vu & Thai, 2020; Huang et al., 2020; Zhang et al., 2021) learns a simple interpretable model for the local neighborhood of a query node and its prediction. The explanations generated by this simple model are treated as the explanations of the original model. We note that BAGEL is, in general, applicable for any explainer which returns binary (hard) or continuous (soft) importance scores (as depicted in Figure 1) for the input features/nodes/edges as an explanation.

## 2.2 Related work on evaluation for post-hoc explanations

Evaluation of explanation methods for any predictive model is inherently tricky. Specifically, when evaluating already trained models, we are faced with the *lack of true explanations*. Collecting true explanations (sometimes referred to as ground truth) for GNNs is even more challenging due to the abstract nature of the input graphs. Moreover, depending on the explanation collection method, it is not always clear if the model used the ground truth in its decision-making process.

Nevertheless, some current works employ small synthetic datasets seeded with a ground truth subgraph. Consequently, metrics such as *explanation accuracy* (Sanchez-Lengeling et al., 2020; Ying et al., 2019) were proposed, which measure the agreement of found explanation with that of ground truth. Observing the false optimism of accuracy metric for small explanation subgraphs, Funke et al. (2022) proposed the use of *Precision* instead of accuracy. We need models that are not only accurate, but right for the right reasons. Towards this, we exploit the text datasets consisting of right reasons or *human rationales* (DeYoung et al., 2019) to construct GNN models. Note that, while being recently popular in the NLP community, comparison with human rationales is missing in the current GNN explanation approaches. To address this gap, we introduce a metric called *plausibility* which measures agreement of explanations with the human rationales. *Plausibility* can be used in conjunction with the *faithfulness* metric which actually evaluates the explainer.

An important notion for evaluating explanations is *faithfulness* where the key idea is to measure how much the explanation characterizes the model's working. To measure faithfulness Sanchez-Lengeling et al. (2020) degrade model performance by damaging the training dataset and measuring how each explanation method responds. The lack of ground truth again limits such a measure. Pope et al. (2019) proposed to compute faithfulness as the difference of accuracy (or predicted probability) between the original predictions and the new predictions after masking out the input features found by the explanation. This was called *Fidelity* in their work. As the features cannot be removed in entirety to measure their impact Funke et al. (2022) proposed RDT-Fidelity based on rate distortion theory defined as the expected predictive score of an explanation over all possible configurations of the non-explanation features.

To measure faithfulness for different explanation types BAGEL uses RDT-Fidelity in addition to two complementary fidelity metrics similar to the one in Pope et al. (2019) and inspired from DeYoung et al. (2019) when only node/edge level explanations are provided.

An important criterion to measure the goodness of an explanation is its size. For example, the full input is also a faithful explanation. However, humans find shorter explanations easier to analyze and reason. Works such as Pope et al. (2019) measure the sparsity of an explanation as the fraction of features selected by the explainer. Noting that this definition is not directly applicable for softmask approaches, Funke et al. (2022) proposes to quantify sparsity as entropy over the normalized distribution of explanation masks. We use the entropy-based sparsity metric as it can be applied both for hard and soft masking approaches.

The authors in Sanchez-Lengeling et al. (2020) argued that the explanation should be stable under input perturbations. In particular, for graph classification, they perturbed test graphs by adding a few nodes/edges such that the final prediction remains the same as that for an unperturbed graph. Lower the change in explanation under perturbations better the stability. A challenge here is that there is no principled way to find the perturbations. For example, a part of the explanation might be altered under random perturbations even if the prediction is unchanged. In the following, we will see that the faithfulness metric of RDT-Fidelity

already accounts for explanation stability without altering the explanation. We also note that there have been other benchmarks to study robustness of GNN models (Fan et al., 2021; Zheng et al., 2021). However, we focus on explaining GNN models predictions and not robustness. Having said this, we affirm that BAGEL could be used in a complementary manner to these existing benchmarks to test trustworthy GNN models.

## 3 Bagel: A Unified Framework for Evaluating Explanations

We now present our framework BAGEL for evaluating GNN explanations. Specifically, BAGEL unifies existing and our proposed notions into 2 main classes. In the *first class* of measures we aim at evaluating the explanation methods in the sense that whether they are truly describing the model's workings. The first category includes three metrics: *faithfulness*, *sparsity* and *correctness*. Faithfulness determines if an explanation alone can replicate the model's behavior. Sparsity focuses on rewarding shorter explanations. Correctness determines if the explanation model is able to detect any injected correlations responsible for altering model's behavior. The metrics in the second class are aimed at evaluating the GNN model itself. Here we propose *plausibility* which measures how close is the decision making process of the trained model (as revealed by explanations) to human rationales.

### 3.1 Faithfulness: Can explanations approximate model's behavior?

The key idea here to evaluate the ability of the explanation to characterize model's working. Unlike previous works we argue that there is not a single measure for faithfulness which can be effectively used for all kinds of datasets and explanations. Consequently we propose a set of two measures to quantify faithfulness depending on the dataset/explanation type.

- RATE DISTORTION BASED FIDELITY. The fidelity of an explanation is usually measured by the ability of an explanation to approximate the model behavior (Ribeiro et al., 2016). For explanations which contain the feature attributions with or without structure attributions, we use the rate distortion theory based metric proposed in Funke et al. (2022) to measure the fidelity of an explanation. In short, a subgraph of the node's computational graph and its set of features are relevant for a classification decision if the expected classifier score remains nearly the same when randomizing the remaining features.

  Let $X$ denotes the input node and features of the computational graph. In particular $X$ corresponds to matrix of nodes in the computational graph and their corresponding feature values. As we use node and feature explanation masks, we compute the final $M(S)$ corresponding to some explanation $S$ by an elementwise product of node and feature masks. The *RDT-Fidelity* of explanation $\mathcal{S}$ respect to the GNN $f$, input $X$ and the noise distribution $\mathcal{N}$ is then given by

  $$\mathcal{F}(\mathcal{S}) = \mathbb{E}_{Y_{\mathcal{S}}|Z \sim \mathcal{N}} \left[ \mathbb{1}_{f(X)=f(Y_{\mathcal{S}})} \right]. \tag{2}$$

  where the perturbed input is given by

  $$Y_{\mathcal{S}} = X \odot M(\mathcal{S}) + Z \odot (\mathbb{1} - M(\mathcal{S})), Z \sim \mathcal{N}, \tag{3}$$

  where $\odot$ denotes an element-wise multiplication, and $\mathbb{1}$ a matrix of ones with the corresponding size and $\mathcal{N}$ is a noise distribution. We choose the noise distribution as the global empirical distribution of the features. We sample the values from the underline training data distribution. The purpose of adding noise is not to replace the unimportant features of input with 0, rather its value should not matter. The replacement of unimportant features with 0 may cause side effects like in some datasets, the value 0 may represent some semantic meaning or biasness towards some pooling strategy, for example, minpool. Also, the noise from global features distribution makes sure that the perturbed data points are still in the same distribution as the original data (Hooker et al., 2019).

  **Connection to explanation stability.** As shown in Funke et al. (2022), explanations with high RDT-fidelity are highly stable. High fidelity score implies that the explanation has high predictive power under perturbations of the rest of the input. Unlike the strategy of Sanchez-Lengeling et al. (2020) to evaluate explanation stability, it is here ensured that the explanation itself is never altered.

**The special case of dense feature representations.** For some datasets it is more appropriate to consider only structure based explanations. For example, when features themselves are dense representations extracted using some black-box embedding method, feature explanations as well as feature perturbations might not make much sense. It is then more appropriate to check the abilities of the explanation with the rest of nodes/edges removed and keeping the features intact. Towards that we employ the following measures of comprehensiveness and sufficiency also used in (DeYoung et al., 2019).

- COMPREHENSIVENESS AND SUFFICIENCY. For explanations which contain only nodes or/and edges we adapt the comprehensiveness and sufficiency measures of DeYoung et al. (2019) for GNNs. Let $\mathcal{G}$ be the graph and $\mathcal{G}' \subseteq \mathcal{G}$ be the explanation graph with important (attribution) nodes/edges. In particular, $\mathcal{G}'$ is generated by removing all nodes/edges from $\mathcal{G}$ which are not part of the explanation.

  Let $f$ be the trained GNN model and $f(\mathcal{G})_j$ be the prediction made by GNN for $j^{th}$ class, where j is the predicted class. We measure fidelity by *comprehensiveness* (which answers the question if all nodes/edges in the graph needed to make a prediction were selected?) and *sufficiency* (if the extracted nodes/edges are sufficient to come up the original prediction?)

$$sufficiency = f\left(\mathcal{G}\right)_j - f\left(\mathcal{G}'\right)_j, \quad comprehensiveness = f\left(\mathcal{G}\right)_j - f\left(\mathcal{G}\backslash\mathcal{G}'\right)_j \tag{4}$$

  A positive value of sufficiency implies that the probability prediction of $f$ on $\mathcal{G}$ is higher than that of $\mathcal{G}'$, which tells us that nodes/edges in the $\mathcal{G}'$ are not sufficient to reach to the same or better prediction. A negative sufficiency score points out that the model $f$ has better prediction on $\mathcal{G}'$ than $\mathcal{G}$ which signifies that explainer was successful in eliminating certain noisy nodes which led to better performance. Similar arguments hold for comprehensiveness. In short these measures should not be symmetric.

  The high *comprehensiveness* value shows that the prediction is most likely because of the explanation $\mathcal{G}'$ and low *comprehensiveness* value shows that $\mathcal{G}'$ is mostly not responsible for the prediction. Since most of the explainers retrieve soft masks we employ aggregated *comprehensiveness* and *sufficiency* measures. In particular, we divide the soft masks into $|\mathcal{B}| = 5$ bins by using top $k \in \mathcal{B} = \{1\%, 5\%, 10\%, 20\%, 50\%\}$ of the explanation with respect to the soft masks values (Samek et al., 2016). The aggregated *sufficiency* is defined as: $\frac{1}{|\mathcal{B}|}\left(\sum_{k=1}^{|\mathcal{B}|} f\left(\mathcal{G}\right)_j - f\left(\mathcal{G}'_k\right)_j\right)$. The aggregated *comprehensiveness* is defined in similar fashion.

  *Remark:* We would like to point out that the subgraph $\mathcal{G}'$ can have disconnected components or even sometimes isolated nodes. The main issue here is that when we convert soft masks to hard masks we might lose the connectivity among the important nodes. It depends on the explainer if it imposed a restriction on returning a connected component. As we are only evaluating the explainer we do not add any additional constraint. Also, since this is a graph classification task, all nodes (from any component) are used towards prediction in the global pooling. For a more holistic evaluation we return the aggregated faithfulness across a set of important explanation subgraphs selected across a range of hard thresholds.

## 3.2 Sparsity: Are the explanations non trivial?

High faithfulness ensures that the explanation approximates the model behavior well. However, the complete input completely determines the model behavior. Thus explanation sparsity is an important criteria for evaluation. Let $p$ be the normalized distribution of explanation (feature) masks. Then sparsity of an explanation is given by the entropy $H(p)$ and is bounded from above by $\log(|M|)$ where $M$ corresponds to a complete set of features or nodes. While an entire input can be a faithful explanation it is important to evaluate an explanation with respect to its size. A shorter explanation is easier to analyse and is more human understandable. We adopt the entropy based definition of sparsity as in Funke et al. (2022) because of its applicability to both soft and hard explanation masks. In particular, let $p$ denote the normalized distribution of node/edge/feature masks. We compute sparsity of an explanation as the entropy over the mask distribution: $H(p) = -\sum_{\phi \in M} p(\phi) \log p(\phi)$.

## 3.3 Correctness: Can the explanations detect externally injected correlations?

While the above measures are essential that the given explanation is predictive certain applications might need explanations for model debugging, for example to detect any spurious correlations picked up by model

thereby increasing model bias. Towards that we measure correctness of an explanation in terms of its ability to recognize the *externally injected correlations* which alters the model decision. A switch in model decision is an evidence of the use of these injected correlations in the actual decision making process.

In particular, we first choose a set of incorrectly labelled nodes, $V$. To each such node $v$, we add edges to the nodes in the training data which have the same label as $v$. We call such edges *decoys*. We retrain the GNN model with the perturbed data. We measure the correctness of explanation $\mathcal{S}$ for nodes in $V$ which are now correctly predicted in terms of precision and recall of the decoys in the returned explanation: $Precision_C = \frac{N_{de}}{N_e}$, $\quad Recall_C = \frac{N_{de}}{N_d}$, where $N_{de}$ is the number of decoys in the obtained explanation, $N_d$ total number of decoys injected and $N_e$ is the size of the retrieved explanation. Note that our proposed approach of using injected correlations is different from using a synthetic graph with seeded ground truth. In particular, for seeded graph approach it is not always clear if the ground truth is actually picked up by the model to make its decision.

### 3.4 Plausibility: How close is the model's decision process to humans rationals?

Human congruence or plausibility (Lei et al., 2016; Lage et al., 2019; Strout et al., 2019) tries to establish how close or congruent is the trained model to human rationales for solving a predictive task. Trained models often exhibit the *clever-hans effect*, that is predictive models can adopt spurious correlations in the training data or due to misplaced inductive biases that have right results for the wrong reasons. Towards this, data is collected from humans for perceptive tasks where humans explicitly provide their rationales. These human rationales are in used as ground truth for evaluating if trained models are right for the right reasons. In Figure 2 we showcase a movie review and the explanations generated (in red) by different explainers. The true label for this review is negative and the GCN makes correct prediction for the review.

| Human Rationales | The first problem that fair game has is the casting of supermodel cindy crawford in the lead role. not that cindy does that bad... sure william is n't a bad actor. unfortunately he just does n't demonstrate it all in this movie... |
|---|---|
| GNNExp | The first problem that fair game has is the casting of supermodel cindy crawford in the lead role. not that cindy does that bad... sure william is n't a bad actor. unfortunately he just does n't demonstrate it all in this movie... |
| Grad | The first problem that fair game has is the casting of supermodel cindy crawford in the lead role. not that cindy does that bad... sure william is n't a bad actor. unfortunately he just does n't demonstrate it all in this movie... |
| CAM | The first problem that fair game has is the casting of supermodel cindy crawford in the lead role. not that cindy does that bad... sure william is n't a bad actor. unfortunately he just does n't demonstrate it all in this movie... |

Figure 2: An anecdotal example of explanations generated by different explainers. The respective plausibility scores for the current example for GNNExp, Grad and CAM are 0.50, 0.54 and 0.61 respectively. We observe that the explanation of CAM agrees best with the human rationales.

For applications where obtaining human rationales is indeed possible we propose the use of *token-level F1* for binary explanation masks and area under precision recall curve (AUPRC) for soft masks. The tokens are words in the input text and are modelled as nodes in the graph. The human rationals are binary masks over the nodes. The token level-F1 score is computed as macro-F1 for predicted binary explanation masks where human rationals serve the true labels. For predicted soft explanation masks we measure additionally the area under precision recall curve. Rather than fixing a threshold, AUPRC provides us a measure of precision -recall tradeoff across different decision thresholds.

The reader might have noticed that this metric is similar to the explanation accuracy in earlier works. We argue against the use of term *accurate* to measure plausibility as similarity to human rationale does not always guarantee that the model has learnt an explanation which contains the reasoning of the model itself and not only of the humans.

## 4  Experimental Setup

**Models and Explainers.** We demonstrate the use and advantage of the proposed framework by evaluating 9 explanation methods over 8 datasets and 4 GNN models. Currently our benchmark consists of these GNN models: graph convolutional networks (GCN) (Kipf & Welling, 2017), graph attention network (GAT) (Veličković et al., 2018), the approximation of personalized propagation of neural predictions (APPNP) (Klicpera et al., 2019), and graph isomorphism network (GIN) (Xu et al., 2019). The models were chosen based on their differences in (i) exploiting inductive biases (based on different feature aggregation strategies), (ii) test performance (see tables 9 and 10 in the Appendix) and (iii) response to injected correlations (see Table 5 and the corresponding discussion). The further details on training of GNNs are available in Appendix M.

We perform experiments with perturbation based approaches like GNNExplainer (GNNExp) (Ying et al., 2019) and Zorro (Funke et al., 2022), surrogate methods like PGM-Explainer (PGM) (Vu & Thai, 2020), and gradient-based approaches like Grad (Simonyan et al., 2013), GradInput(Simonyan et al., 2013), Integrated Gradient (IG) (Sundararajan et al., 2017), SmoothGrad (Smilkov et al., 2017), CAM (Pope et al., 2019) and GradCAM (Pope et al., 2019). GNNExp returns soft feature masks and edge masks. We transform the edge masks into node masks, in which we equally distribute the edge importance score to both nodes sharing the edge. The further details of these explainers are available in Appendix B. As already mentioned BAGEL is extendable, and more approaches and explainers can be easily added.

### 4.1  Datasets

We now describe new and existing datasets used in our evaluation framework and the corresponding rationale.

**New Dataset for Plausibility.** To measure the plausibility of an explanation, we first require the corresponding human rationales. Since the existing graph datasets do not have such annotated information, we transform a text sentiment prediction task into a graph classification task. Specifically, we adopt the Movie Reviews dataset (Zaidan & Eisner, 2008) from the ERASER benchmark (DeYoung et al., 2019). The task is the binary classification, which is the differentiation between positive and negative movie reviews.

**Construction of Movie Reviews Dataset**. Each input instance or review is a passage of text, typically with multiple sentences. Each input review is annotated by humans that reflects the actual "human" reasons for predicting the sentiment of the review. These annotations are are extractive pieces of text and we call them human rationales. We transform sentences into graphs using the graph-of-words approach (Rousseau & Vazirgiannis, 2013). As a pre-processing step, we remove stopwords, such as "the" or "a". The complete list of used stopwords is included in our repository. Each word is represented as a node and all words within a sliding window of three are connected via edges. As features, we use the output of a pre-trained Glove model (Pennington et al., 2014). Figure 3 provides an example of a graph from Movie Reviews dataset.

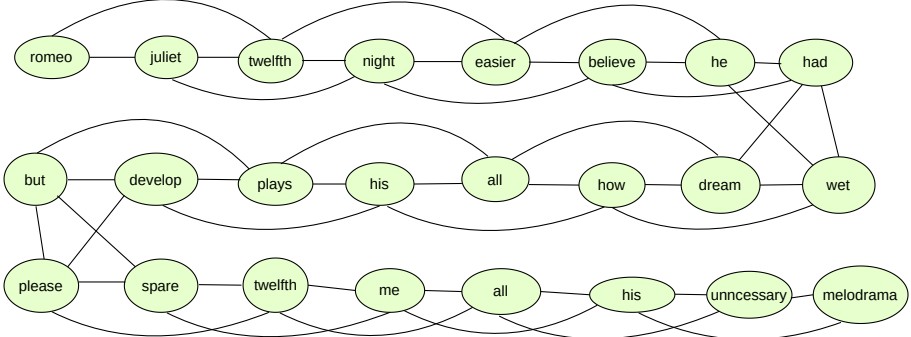

Figure 3: An example for text to graph generation. The graph is corresponding to input sentences *"? romeo and juliet ' , and ? the twelfth night ' . it is easier for me to believe that he had a wet dream and that 's how all his plays develop , but please spare me all of this unnecessary melodrama."*

**Dataset to measure Comprehensiveness and Sufficiency.** Note that the measures comprehensiveness and sufficiency are only applicable for node/edge explanations. We use the above-described Movie Review dataset for these two measures too. The rationale is that the node features are generated using Glove and are not human-understandable. In this case, a structure-based explanation would be more meaningful than a feature-based one. Further, we evaluate comprehensiveness and sufficiency on MUTAG, PROTEINS and ENZYMES datasets.

**Datasets for Correctness.** We employ two citation datasets Cora (Sen et al., 2008) and CiteSeer (Sen et al., 2008). After injecting correlations/decoys corresponding to incorrectly labeled nodes as described in Section 3.3, we re-train the GNN model. The rationale behind adding homophily increasing correlations is the observation from previous works (Khosla et al., 2019; Zhu et al., 2020) that GNN's performance increases with higher homophily. A model will have picked up these correlations if the previous incorrect nodes are now correctly predicted. We further evaluate the correctness of the explanation only for newly correctly predicted nodes.

**Datasets for RDT-Fidelity.** We perform the *RTD-Fidelity* evaluation on both *node classification* and *graph classification* tasks. At node level, we use Citation datasets namely Cora, CiteSeer, PubMed, ogbn-arxiv (Hu et al., 2020a) and synthetic datasets Ying et al. (2019). Table 9 shows the dataset statistics and GNNs performances. For node classification task, we select 300 nodes for Cora and CiteSeer, and PubMed and 1000 nodes for ogbn-arxiv randomly. We choose a smaller sample of nodes to get explanations due to longer run times of certain explainers. The sample is chosen randomly to avoid any biases. Moreover, in Figure 11, we provide additionally the standard deviation for explainer performance corresponding to node samples. For graph classification task, we use MUTAG (Debnath et al., 1991), PROTEINS (Borgwardt et al., 2005) and ENZYMES Morris et al. (2020) datasets. We select 50 graphs for both MUTAG and PROTEINS datasets and 200 graphs for ENZYMES dataset.

## 5 Result Analysis

### 5.1 Faithfulness

Table 2: Results for *RDT-Fidelity* for node classification.

| Mask | Methods | Cora | | | | CiteSeer | | | | PubMed | | | |
|---|---|---|---|---|---|---|---|---|---|---|---|---|---|
| | | GCN | GAT | GIN | APPNP | GCN | GAT | GIN | APPNP | GCN | GAT | GIN | APPNP |
| Hard | Zorro | **0.97** | **0.97** | **0.96** | **0.97** | **0.97** | **0.97** | **0.97** | **0.96** | **0.96** | **0.97** | **0.97** | **0.96** |
| | PGM | 0.84 | 0.77 | 0.60 | 0.89 | 0.92 | 0.93 | 0.73 | 0.95 | 0.78 | 0.69 | 0.74 | 0.96 |
| Soft | GNNExp | 0.71 | 0.66 | 0.52 | 0.65 | 0.68 | 0.69 | 0.51 | 0.62 | 0.67 | 0.73 | 0.67 | 0.72 |
| | Grad | 0.15 | 0.18 | 0.19 | 0.17 | 0.17 | 0.19 | 0.28 | 0.18 | 0.37 | 0.43 | 0.42 | 0.37 |
| | GradInput | 0.15 | 0.18 | 0.18 | 0.16 | 0.16 | 0.18 | 0.26 | 0.17 | 0.36 | 0.42 | 0.42 | 0.36 |
| | SmoothGrad | 0.44 | 0.42 | 0.38 | 0.50 | 0.54 | 0.57 | 0.45 | 0.62 | 0.52 | 0.53 | 0.67 | 0.59 |
| | IG | 0.45 | 0.47 | 0.26 | 0.51 | 0.53 | 0.70 | 0.45 | 0.62 | 0.52 | 0.56 | 0.68 | 0.59 |
| | Empty Expl. | 0.15 | 0.18 | 0.18 | 0.16 | 0.16 | 0.18 | 0.26 | 0.17 | 0.36 | 0.42 | 0.42 | 0.36 |
| | Random Expl. | 0.63 | 0.60 | 0.42 | 0.55 | 0.59 | 0.57 | 0.52 | 0.52 | 0.75 | 0.67 | 0.67 | 0.70 |

**RDT-Fidelity.** In Table 2 we compare the RDT-Fidelity scores of various explanation methods. A common feature of Zorro and PGM is that they both learn the explanations from a sampled local dataset. The local dataset is created by perturbing the features of nodes from the computational graph (neighborhood of query nodes). While they employ different optimization strategies to find explanations, the result is a stable explanation that also reflects our results. The gradient-based explanations achieve the lowest fidelity. We also choose empty and random explanations as baselines. In an empty explanation, the importance scores for all nodes and features are set to be 0. In case of a random explanation, we select nodes/features masks randomly from a uniform distribution. We observe that empty explanation performs similar to GradInput. The reason for this is that the explanation mask output by GradInput is close to a zero vector.

Table 3: Faithfulness as *comprehensiveness* and *sufficiency* measured for Movie Reviews dataset. Low sufficiency and high comprehensiveness indicates high faithfulness.

| Methods | GCN | | GAT | | GIN | | APPNP | |
|---------|------|------|------|------|------|------|------|------|
| | Suff. | Comp. | Suff. | Comp. | Suff. | Comp. | Suff. | Comp. |
| GNNExp | 0.56 | -0.01 | 0.47 | 0.03 | 0.24 | 0.32 | 0.48 | 0.07 |
| Grad | **0.02** | **0.39** | 0.15 | 0.16 | 0.25 | 0.31 | 0.11 | 0.20 |
| GradInput | 0.07 | 0.36 | 0.14 | 0.16 | 0.22 | **0.33** | 0.12 | 0.24 |
| SmoothGrad | 0.08 | 0.34 | 0.15 | 0.23 | 0.25 | 0.27 | 0.14 | 0.23 |
| IG | 0.11 | 0.38 | 0.16 | 0.21 | 0.23 | 0.27 | 0.16 | 0.24 |
| CAM | 0.41 | 0.01 | 0.11 | 0.14 | 0.26 | 0.26 | 0.39 | 0.05 |
| GradCAM | **0.02** | 0.29 | **0.06** | **0.27** | **0.21** | 0.27 | **0.07** | **0.28** |

For the graph classification task (see Table 11 in Appendix), all methods, including the gradient-based approaches, perform relatively well except for the GCN model. PGM shows more consistent performance across all models and datasets. We leave out Zorro as it is not applicable for the graph classification task.

**Comprehensiveness and Sufficiency.** We evaluate faithfulness for explanations for the Movie Reviews dataset using aggregated *comprehensiveness* and *sufficiency* measures. The results for soft-mask explanations are shown in Table 3. GradCAM has the lowest *sufficiency* which suggests that the explanations are sufficient to mimic the prediction of GNN models. On the other hand, the explanations generated by GradCAM with GCN and GIN suggest that there still exists important part of the input outside of the explanations which are required to approximate the GNN's prediction. On the other hand, GNNExp, which so far outperformed gradient-based explanations for the node classification task, shows the worst sufficiency and comprehensiveness. Even if we use the complete feature set and only the node masks to evaluate explanations, the node masks for GNNExp are learned together with the feature explanations. This differs from gradient-based approaches, which ignore feature and structure explanation trade-offs. The current performance of GNNExp indicates that it might not be appropriate to use entangled features and structure explanations independently. We report the *comprehensiveness* and *sufficiency* on the MUTAG, PROTEINS and ENZYMES dataset in Appendix D (See Tables 13 to 16). We observe that there is no single explainer which outperforms consistently with all GNNs on these 3 molecule datasets. In Table 16, we report *sufficiency* and *comprehensiveness* of molecules datasets trained with GCN model. We calculate the *sufficiency* and *comprehensiveness* when the edge masks are used to generate induced subgraphs with different thresholds. GNNExp-Edge represents GNNExp when edge masks are directly used as the explanations. GradEdge represents gradients over edges. We also use random edge masks as a baseline. GNNExp-Edge outperforms on MUTAG and ENZYMES datasets and GradEdge outperforms on PROTEINS dataset.

## 5.2 Sparsity

As already mentioned, a complete input could already do well on all faithfulness measures. Therefore, we further look for sparser explanations. The results for node sparsity for explanations in node classification task are provided in Table 4. For the hard masking approaches (Zorro and PGM), Zorro outperforms PGM with all GNN models except for GIN. Conversely, there is no clear winner for the soft mask approach. The high sparsity for soft-masking approaches implies a near-uniform node attribution and consequently lower interpretability. In general, faithfulness and sparsity of an explanation should be analyzed together. A uniformly distributed explanation mask could already provide an explanation with high faithfulness as it leads to using the complete input as an explanation. We also report feature sparsity for node classification in Appendix E (in Table 17). We observe similar trends on features level sparsity where Zorro outperforms over almost all datasets except SmothGrad when GIN is trained on CITESEER. We further report node sparsity on MUTAG, PROTEINS and ENZYMES in Appendix C (in table 12), where PGM outperforms over all three datasets.

## 5.3 Correctness

Table 4: Results for sparsity (computed as entropy over mask distribution) for node classification. The lower the score sparser is the explanation.

| Mask | Methods | Cora | | | | CiteSeer | | | | PubMed | | | |
|------|---------|------|------|------|-------|------|------|------|-------|------|------|------|-------|
| | | GCN | GAT | GIN | APPNP | GCN | GAT | GIN | APPNP | GCN | GAT | GIN | APPNP |
| Hard | Zorro | **1.58** | **1.59** | 2.17 | **1.48** | **1.26** | **1.09** | 1.58 | **1.07** | **1.51** | **1.31** | 2.18 | **1.25** |
| | PGM | 2.06 | 1.82 | **1.66** | 1.99 | 1.47 | 1.59 | **1.10** | 1.54 | 1.64 | 1.16 | **1.62** | 2.93 |
| Soft | GNNExp | 2.48 | 2.49 | 2.56 | 2.51 | 1.67 | 1.67 | 1.70 | 1.68 | 2.70 | 2.71 | 2.71 | 2.71 |
| | Grad | 2.48 | 2.34 | 2.25 | 2.35 | 1.70 | 1.61 | 1.55 | 1.60 | 2.91 | 2.76 | 3.11 | 2.73 |
| | GradInput | 2.53 | 2.43 | 2.23 | 2.41 | 1.61 | 1.58 | 1.54 | 1.52 | 3.02 | 2.94 | 3.41 | 2.81 |
| | SmoothGrad | 2.48 | 2.52 | 2.91 | 2.31 | 1.77 | 1.77 | 1.93 | 1.66 | 2.89 | 3.02 | 3.23 | 2.54 |
| | IG | 2.49 | 2.50 | 2.84 | 2.31 | 1.76 | 1.77 | 1.91 | 1.66 | 2.84 | 2.89 | 3.06 | 2.58 |
| | Random Expl. | 7.71 | 7.71 | 7.71 | 7.71 | 7.92 | 7.92 | 7.92 | 7.92 | 9.69 | 9.69 | 9.69 | 9.69 |

The correctness results corresponding to different models and explainers are reported for Cora (in Table 6) and CiteSeer (in Table 18). We report precision, recall and F1 score by choosing top k nodes for the soft explanations. For hard masked approaches the number of returned nodes is listed under $|\mathcal{S}|$. Note that the number of decoys added per node is 10. For Table 6 and Table 18 we use $k = 20$.

In Table 5, the effect of decoys can be seen where most of the earlier incorrectly classified nodes are now correctly classified except for GCN on CiteSeer. We also observe that number of selected nodes for GIN is very low for Cora dataset (i.e., only a few nodes were initially incorrectly labeled). GNNExp outperforms all other based explainers in detecting the injected correlations for both Cora and CiteSeer (detailed results moved to Table 18 in the Appendix due to space constraints).

Table 5: The number of incorrectly labelled nodes (✗) decreases after addition of decoys. The number of new correctly labelled nodes after injecting decoys is listed under ✓.

| Model | Cora | | | CiteSeer | | |
|-------|------|------|------|----------|------|------|
| | ✗ | ✓ | ↑(%) | ✗ | ✓ | ↑(%) |
| GCN | 88 | 79 | 89.7 | 329 | 229 | 69.6 |
| GAT | 86 | 85 | 98.8 | 311 | 301 | 96.7 |
| GIN | 6 | 6 | 100 | 56 | 56 | 100 |
| APPNP | 73 | 70 | 95.8 | 280 | 252 | 90.0 |

Comparing soft mask and hard mask approaches in this setting is tricky as for some approaches like Zorro, we cannot control the explanation size. For example, for GAT Zorro retrieved an explanation of size 40. A precision of 0.25 shows that it found all 10 injected correlations. Lack of feature ranking, as in soft mask approaches, makes it difficult to evaluate hard mask approaches for Correctness. For fairer evaluation, we further plot the performance of soft mask approaches with different $k$ in Appendix H. For example, the GNNExp shows high improvement when we increase the size of the explanation to 15. It is not surprising to see the performance degrades when we increase the size of the explanation further since it already had captured all injected decoys. Now it returns some irrelevant nodes in the explanation. Furthermore, in Table 20 and 21, we use mean as a threshold to generate hard masks. As the mean threshold turns out to be very low for all approaches, almost all nodes of the computational graph are selected as the explanation. Consequently, we observe a very low correctness score (when measured in terms of precision).

### 5.4 Plausibility

Table 7 shows the *Plausibility* scores computed for explanations of different GNN models. Recall that we compare explanations with human rationales to compute plausibility. The average size of human rationales over the test dataset is 165. To compute token level F1 score, we use mean as a threshold to generate hard masks from soft masks.

We observe that all explainers assign the best plausibility scores to GCN. GIN obtains the second-best plausibility scores. We also observe that the overall difference in the plausibility scores over models is relatively small, with some exceptions like the combination of GIN and GNNExp. The corresponding explanation also has the largest size. This further highlights the issues of soft-hard mask conversion. AUPRC scores which directly use the soft masks are more stable. One surprising fact in these results is that even though other GNN models achieve higher test accuracy than GIN (see Table 10 in the Appendix). Overall, their

Table 6: Correctness of the explanation for node classification on CORA dataset. We use $k = 20$.

| Mask | Methods | CORA | | | | | | | | | | | | | | | |
| | | GCN | | | | GAT | | | | GIN | | | | APPNP | | | |
| | | P@k | R@k | F1 | $|\mathcal{S}|$ | P@k | R@k | F1 | $|\mathcal{S}|$ | P@k | R@k | F1 | $|\mathcal{S}|$ | P@k | R@k | F1 | $|\mathcal{S}|$ |
| Hard | Zorro | 0.19 | 0.80 | 0.30 | 45 | 0.25 | 0.83 | 0.37 | 40 | 0.26 | 0.45 | 0.27 | 33 | 0.22 | 0.79 | 0.33 | 38 |
| | PGM | 0.11 | 0.22 | 0.15 | 20 | 0.18 | 0.36 | 0.24 | 20 | 0.18 | 0.36 | 0.25 | 20 | 0.19 | 0.38 | 0.25 | 20 |
| Soft | GNNExp | **0.42** | **0.84** | **0.56** | 20 | **0.44** | **0.88** | **0.59** | 20 | **0.50** | **1.00** | **0.67** | 20 | **0.34** | **0.67** | **0.58** | 20 |
| | Grad | 0.23 | 0.46 | 0.31 | 20 | 0.29 | 0.58 | 0.39 | 20 | 0.30 | 0.60 | 0.40 | 20 | 0.33 | 0.67 | 0.45 | 20 |
| | GradInput | 0.16 | 0.32 | 0.21 | 20 | 0.28 | 0.56 | 0.34 | 20 | 0.30 | 0.60 | 0.40 | 20 | 0.28 | 0.56 | 0.38 | 20 |
| | SmoothGrad | 0.12 | 0.25 | 0.16 | 20 | 0.24 | 0.48 | 0.32 | 20 | **0.50** | **1.00** | **0.67** | 20 | 0.22 | 0.43 | 0.29 | 20 |
| | IG | 0.16 | 0.32 | 0.22 | 20 | 0.24 | 0.49 | 0.33 | 20 | **0.50** | **1.00** | **0.67** | 20 | 0.28 | 0.55 | 0.37 | 20 |

explanations have similar plausibility as for GIN except for GNNExp. In such cases, an application user might want to look in more detail at specific correctly labeled instances to check if the model imitates human reasoning.

We now provide a concrete example of how plausibility metric can be used in conjunction with faithfulness to metric to evaluate model's decision making process. From our previous example in Figure 2 we choose Grad which achieves the best faithfulness score on this example. In Figure 4 we compare the explanations of different models as provided by Grad explainer and compare the explanations based on plausibility. We observe the for GCN it achieves the highest faithfulness and plausibility scores.

Table 7: Plausibility for movie review dataset measured by auprc and F1 score (macro). $|\mathcal{S}|$ represents the average size of the explanations generated by the explainers.

| Mask | Methods | GCN | | | GAT | | | GIN | | | APPNP | | |
| | | auprc | F1 | $|\mathcal{S}|$ | auprc | F1 | $|\mathcal{S}|$ | auprc | F1 | $|\mathcal{S}|$ | auprc | F1 | $|\mathcal{S}|$ |
| Hard | PGM | — | 0.42 | 25 | — | **0.43** | 25 | — | **0.43** | 25 | — | **0.43** | 25 |
| Soft | GNNExp | 0.46 | **0.54** | 168 | 0.43 | **0.54** | 149 | 0.45 | 0.35 | 410 | 0.45 | 0.53 | 158 |
| | Grad | 0.44 | **0.52** | 265 | 0.38 | 0.51 | 158 | 0.40 | **0.52** | 156 | 0.38 | 0.50 | 255 |
| | GradInput | 0.39 | 0.51 | 221 | 0.37 | 0.50 | 154 | 0.39 | **0.51** | 154 | 0.37 | 0.50 | 227 |
| | SmoothGrad | 0.40 | **0.52** | 219 | 0.37 | 0.50 | 154 | 0.40 | **0.52** | 172 | 0.38 | 0.50 | 221 |
| | IG | 0.37 | 0.49 | 225 | 0.37 | 0.50 | 188 | 0.39 | **0.51** | 186 | 0.38 | 0.50 | 219 |
| | CAM | 0.54 | **0.61** | 224 | 0.40 | 0.51 | 177 | 0.44 | 0.55 | 156 | 0.44 | 0.53 | 195 |
| | GradCAM | 0.67 | 0.34 | 175 | 0.67 | **0.35** | 191 | 0.67 | 0.34 | 166 | 0.67 | 0.34 | 188 |

| | |
|---|---|
| GCN | The first problem that fair game has is the casting of supermodel cindy crawford in the lead role. not that cindy does that bad... sure william is n't a bad actor. unfortunately he just does n't demonstrate it all in this movie... |
| GAT | The first problem that fair game has is the casting of supermodel cindy crawford in the lead role. not that cindy does that bad... sure william is n't a bad actor. unfortunately he just does n't demonstrate it all in this movie... |
| APPNP | The first problem that fair game has is the casting of supermodel cindy crawford in the lead role. not that cindy does that bad... sure william is n't a bad actor. unfortunately he just does n't demonstrate it all in this movie... |

Figure 4: An example to illustrate the use of *plausibility* in conjunction with *faithfulness* to select the model that best agrees with human rationales. We compare different models for Grad explanations because Grad explanations are highly faithful. Grad explanations over GCN agree best with human rationales.

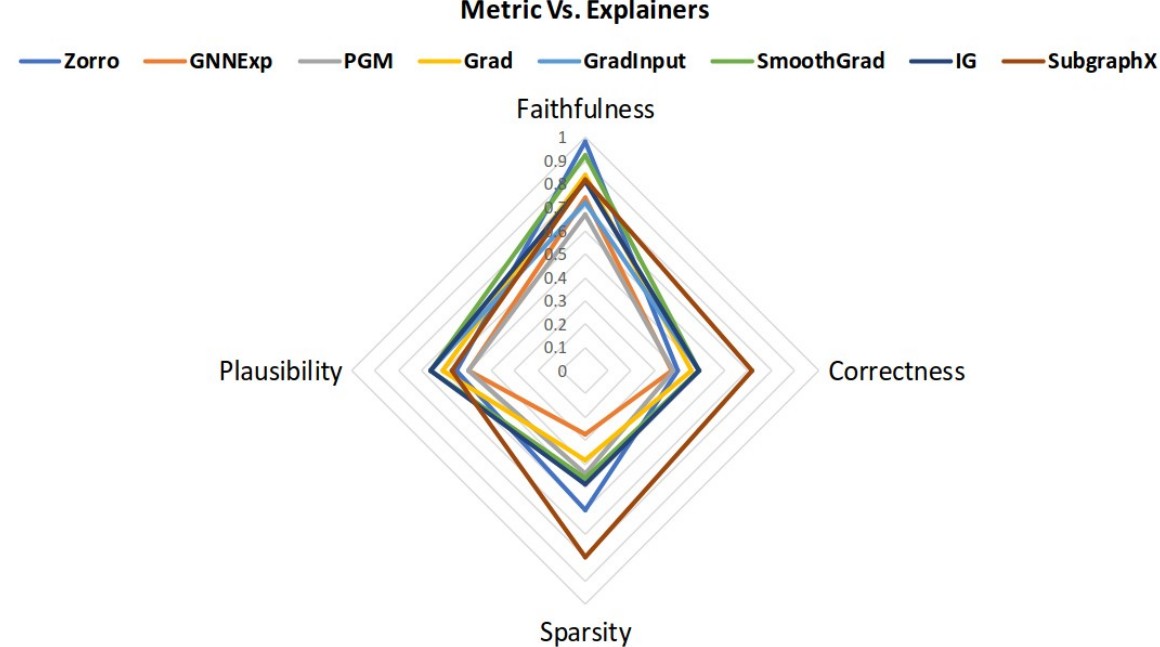

Figure 5: An overview of performances on Synthetic dataset. For each metric, the higher the better.

### 5.5 All Metrics on Synthetic dataset

In Table 8, we report performances over all metrics on synthetic dataset using GCN. We use synthetic dataset proposed by Ying et al. (2019), where five-node house graph is attached to randomly selected nodes, which works as the ground truth. Since the ground truths are the reasons for the prediction by the design of dataset construction, we treat the ground truth nodes as decoys to compute correctness. The further details on dataset are available in Appendix A. We train a 3-layers GCN model on synthetic dataset. We evaluate all metrics on 700 randomly selected nodes. We added SubgraphX Yuan et al. (2021) for this experiment. For hard masks approaches Zorro and SubgraphX, we can not retrieve the explanations of pre-defined size. For PGM and all soft mask approaches, we choose top k nodes as the explanation. We use $k = 10$. Similar to real-world datasets, Zorro generates the most faithful explanations for synthetic dataset. For gradient based approaches, SmoothGrad generates most faithful explanations. The low sparsity and high precision of explanations show that the explanations generated by SubgraphX are small and correct as well. Further we observe that the gradient based approaches like GradInput, SmoothGrad and IG assign the high plausible score to GCN. In Figure 5, we plot the all four metrics on Synthetic dataset. Since for the used sparsity metric, lower is better, we use reciprocal of original sparsity in the plot.

## 6  Conclusion

We develop a unified, modular, extendable benchmark called BAGEL to evaluate GNN explanations on four diverse axes: 1) *faithfulness*, 2) *sparsity*, 3) *correctness*, and 4) *plausibility*. Faithfulness measured via *RDT-Fidelity* can be employed for a wide set of tasks and datasets. We note that high RDT-Fidelity also implies high explanation stability. The *comprehensiveness* and *sufficiency* measures should be used to evaluate the faithfulness of structure-based explanations where perturbing features might not be feasible. It is important to measure the sparsity of the explanation to avoid the extreme case of using the whole input as an explanation. Correctness should be used with care, as injecting appropriate correlations to change a model's decision is not always straightforward.

Table 8: All metrics on synthetic dataset. We use $k = 10$.

| Mask | Methods | *Faithfulness* RDT-Fidelity | Sparsity | Correctness P@k | R@k | Plausibility auprc | F1 |
|------|---------|------|------|------|------|------|------|
| Hard | Zorro | **0.98** | 1.68 | 0.40 | 0.69 | na | 0.55 |
|      | PGM | 0.67 | 2.28 | 0.38 | 0.73 | na | 0.50 |
|      | SubgraphX | 0.82 | **1.25** | **0.72** | 0.53 | na | 0.57 |
| Soft | GNNExp | 0.74 | 3.65 | 0.37 | 0.75 | 0.56 | 0.50 |
|      | Grad | 0.84 | 2.60 | 0.46 | 0.92 | 0.69 | 0.61 |
|      | GradInput | 0.72 | 2.13 | 0.49 | **0.99** | **0.74** | **0.66** |
|      | SmoothGrad | 0.92 | 2.17 | 0.49 | **0.99** | **0.74** | **0.66** |
|      | IG | 0.81 | 2.06 | 0.49 | **0.99** | **0.74** | **0.66** |
|      | Zero Exp. | 0.50 | 0.00 | 0.00 | 0.00 | 0.00 | 0.00 |
|      | Random Exp. | 0.83 | 7.05 | 0.01 | 0.01 | 0.01 | 0.01 |

Plausibility measures the joint utility of the explanation method and the trained GNN model with respect to human rationales. Assuming that the generated explanations are faithful to the model, one can use plausibility to check the model's congruence to human rationales. This means that the loss of plausibility can be either due to human-incongruent correlations or due to non-faithfulness of the explainer. To fully interpret the results of plausibility one should first check explanation faithfulness.

**Broader Impact Statement**

By providing a unified evaluation framework we hope to have a positive impact on the further development and holistic evaluation of explainability techniques for graph neural networks. Given the growing applications of GNNs in various sensitive domains, such assessment benchmarks are essential to provide different perspectives on the provided explanation. On the other hand, automatic deployment of GNN explanations have been shown to lead to leakage of data privacy (Olatunji et al., 2022). A more general perspective on privacy-transparency tradeoffs in graph machine learning is provided in Khosla (2022).

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
