# OpenReview forum: "Bagel: A Benchmark for Assessing Graph Neural Network Explanations"
_TMLR — Rejected by TMLR_

### Review · Reviewer_fAW4 · 2022-12-06

**Summary Of Contributions:**

This work presents a benchmark for evaluating graph neural network explanations. The benchmark contains multiple metrics and datasets. Most metrics are from existing works. But the paper seems to propose a new metric named “correctness” which checks whether the explanations can detect externally injected correlations.

**Audience:**

Yes

**Claims And Evidence:**

No

**Requested Changes:**

- [Critical] Include multiple different datasets for every metric.
- [Critical] Justify the new “correctness” metric.


**Strengths And Weaknesses:**

Strengths:
- This work provides a benchmark which may help future works to evaluate explanation methods.
- The work constructed a Movie Reviews dataset using NLP to construct graphs which have corresponding human rationales for measuring plausibility.

Weaknesses:
- The contribution is limited as a significant part of this paper is about combining existing metrics and datasets.
- For a benchmark work, this paper is not comprehensive enough to contain multiple different datasets for every metric. Metrics such as comprehensiveness, sufficiency, plausibility are only evaluated on the single Movie Reviews dataset created in this work.
- A new “correctness” metric is proposed, but the soundness of the new metric is not justified by extensive experiments or theory, i.e., why is this a good metric and why is having a high “correctness” desired for explanation methods.

----Post rebuttal updates----

Thanks to the authors for the repsonse. Constructing the graph classification dataset from the text classification task is indeed a contribution of this paper. But I am not quite convinced about the soundness of plausibility and correctness:

- Plausibility: I don't think a good interpretation should match human rationales, as the interpretation should faithfully interpret the behavior of the model which may make inference in a totally different way compared to humans;
- Correctness: In this metric, the paper mentions "We retrain the GNN model with the perturbed data". I think it is hard to tell if a good interpretation should identify the decoys, if the models are retrained, which is a complex process. The authors provided some explanation in the author response, but the soundness of the metric is still not justified by experiments or theory.

I think the paper may be improved to better justify the soundness of the metrics or propose more sound metrics.

---

> ### Author Response · Authors · 2022-12-20
> **Novelty explained, more experiments added**
>
> Thank you for your constructive comments. In the following we provide a few clarifications and details about changes made in the paper. We are enthusiastically looking forward to any further discussion.
> > The contribution is limited as a significant part of this paper is about combining existing metrics and datasets.
>
> In choosing relevant metrics for the benchmark, we firstly started with the important metrics already proposed in the literature and categorized them based on their function. In this sense, we re-used existing metrics. However, the plausibility metric is our contribution. Also novel are graph classification datasets constructed from a text classification task, i.e., the movie reviews (MR) dataset. We note that converting them into graphs and considering it as a graph classification task for evaluating GNN explanations is indeed new. In addition, for measuring correctness, we propose the experimental protocols of adding decoys are also arguably novel and our contributions. Further we evaluate the explainers on (majority) real world datasets under one umbrella.
>
> >For a benchmark work, this paper is not comprehensive enough to contain multiple different datasets for every metric. Metrics such as comprehensiveness, sufficiency, plausibility are only evaluated on the single Movie Reviews dataset created in this work.
>
> Thanks for the constructive comments. We have added more experiments in our revised version. Particularly, we evaluate sufficiency & comprehensiveness, and Sparsity on MUTAG, PROTEINS, and ENZYMES(Table 13, 14, 15 and 12 respectively). We added feature sparsity on node classification datasets (Cora, CiteSeer and PubMed) in Table 17. We evaluate Plausibility, RDT-fidelity, and Sparsity on Synthetic datasets which for ground truth (human rationale) is available (In Table 8).  We have added Table 1, which contains information which dataset is used for which metric and tasks.
> Further we would like to point out that in practice, the evaluation of all 4 regimes on a single dataset is not always possible. Different datasets have different notions of interpretability. For example the RDT-fidelity makes sense when we have sparse features (each feature contains some unique information about the input) and we perturb the features based on the explanation keeping the features distribution same. On the other hand in the text dataset, the features are dense and the perturbation over features does not make sense. Therefore, our benchmark unifies all metrics into these four categories with a representative task, metric, and dataset for each category.
> Our benchmark is very easy to extend where we are continuously adding new datasets and explainers.
>
> > A new “correctness” metric is proposed, but the soundness of the new metric is not justified by extensive experiments or theory, i.e., why is this a good metric and why is having a high “correctness” desired for explanation methods.
>
> In correctness, we want to see if the explanations are indeed aligned by the underlined GNN models. However we do not know if the actual reasons or patterns are used by the machine towards making predictions. For example, the model predicts the right label after injecting external decoys, (where decoys are the reason for the prediction), now we want the explainer to pick these decoys in the explanation.
>
> **Why is correctness a good metric?**
>
> * The added decoys work as ground truth, which is not present for the most of the graph datasets.
> * The ability of detecting externally injected decoys of explainers can be used in model debugging.
>
> **Why is Correctness different from RDT-Fidelity and Plausibility?**
>
> RDT-Fidelity measures if the explanation can mimic the model’s prediction over perturbation. For which we do not need ground truth. Plausibility measures how close the decision making process of the trained model is to human rationales. But in correctness we measure if the explanation can detect the externally injected decoys.

---

> ### Author Response · Authors · 2023-01-09
> **response to post rebuttal updates**
>
> Thank you for your comments. In the following we provide further clarifications on Plausibility and Correctness. We are enthusiastically looking forward to any further discussion.
>
> > Plausibility: I don't think a good interpretation should match human rationales, as the interpretation should faithfully interpret the behavior of the model which may make inference in a totally different way compared to humans;
>
> As already argued in the paper, plausibility should always be used together with the faithfulness metric. Plausibility can be used to answer the following question “ Among several faithful explanations which one is the most human understandable”?
>
> It is true that models may make predictions different from how humans think, but the alignment of explanations to the human rationales increases trust between humans and the decision process of models. Further the explanations that match human rationales are often easier for humans to understand and interpret. Finally, plausibility is easiest and cleanest to evaluate since they are typically grounded in human perception. Due to this many lines of work in interpretability in text and vision (but missing in graphs) use plausibility as an important aspect for interpretability [1], and [2]. Since there was no previous real-world dataset evaluating plausibility in graphs, our contribution is to extend a plausibility setting for measuring human congruence of explanations in GNNs. We hope that this context helps, and we would be willing to make it more explicit in the text.
>
> >Correctness: In this metric, the paper mentions "We retrain the GNN model with the perturbed data". I think it is hard to tell if a good interpretation should identify the decoys, if the models are retrained, which is a complex process. The authors provided some explanation in the author response, but the soundness of the metric is still not justified by experiments or theory.
>
> In the particular case of correctness considered in the paper, we add the decoys to increase the homophily for incorrectly classified nodes. To make sure that these newly added decoys are responsible for label flip (from incorrect to correct), we re-train the model and evaluate the correctness only for newly correctly classified nodes (out of incorrectly classified nodes). We report the % of correctly classified nodes after adding decoys in Table 5. The change in decision for the nodes for whom the decoys are added shows that the decoys were used for decision making. More explicitly, we operate under an assumption that if the model has “learnt” the decoy pattern, it should be reflected in the model performance. Please note that this important but overlooked “verification” step is missing in most of the decoy literature as pointed out by [3]. So we respectfully disagree with the claim that this protocol is not based on proper experimentation. In fact, we would like to think that we have taken a rigorous control setting for establishing correctness through decoys using an important verification step that is grounded and accepted in empirical studies.
>
> **Why should the explainer pick the decoys in the explanation?**
>
> This is evident from the change in predicted labels of nodes for whom the decoys were added. Now the model has picked the decoys and altered its decision, to make sure that the prediction made by the model is right for the right reasons (where decoys work as ground truth), we want the explainers to pick these decoys in the explanation.
>
> **Why is re-training necessary?**
>
> As we know, a 2-layer GNN model takes 2-hop neighbors into consideration in making the prediction and hence a good explainer tries to retrieve only those neighbors which are highly responsible for making decisions i.e., the nodes which GNN used in making decisions. All the explainers take trained GNN model and graph as input and generate explanations. Further [4] proposed that adding noise to the dataset may change the distribution (nodes’ degrees in our scenario ) of the original dataset on which the model was trained. To interpret the model’s decision over the perturbed dataset, the model should be re-trained.
>
>
> [1] ERASER : A Benchmark to Evaluate Rationalized NLP Models, Jay DeYoung et al., 2020
>
> [2] Right for the right reasons: Training differentiable models by constraining their explanations. Ross, Andrew Slavin, Michael C. Hughes, and Finale Doshi-Velez, 2017.
>
> [3]  Towards Benchmarking the Utility of Explanations for Model Debugging, Maximilian Idahl et al. 2021
>
> [4] A benchmark for interpretability methods in deep neural networks. Sara Hooker, Dumitru Erhan, Pieter-Jan Kindermans, and Been Kim. 2019

---

### Review · Reviewer_J5LF · 2022-12-06

**Summary Of Contributions:**

The authors provide a framework how to evaluate the explanation of graph neural network. They provide metrics to check whether a GNN explanation is faithful, sparse, correct and/or plausible. They apply their metrics using different datasets, models and explanation methods.

**Audience:**

Yes

**Broader Impact Concerns:**

The paper doesn't seem to be integrated sufficiently in the existing research landscape, since many references are missing. In addition, as said above, the actual contribution seems quite mild or not sufficiently discussed.

**Claims And Evidence:**

Yes

**Requested Changes:**

My requested changes are provided with the description of the weaknesses, above.

A minor change: I think in the first line of page 4 you mean \mathcal{G}' and \mathcal{G} rather than just G' and G.



**Strengths And Weaknesses:**

pros:
 The paper is clearly written and discusses a very important subject. The authors provide a good and intuitive experimental setup where different properties of different explanation methods are provided.

cons:
- The references are not complete. The paper is about evaluating explanation methods for GNNs, yet only very few GNN explanation methods are provided. In my opinion the authors only use and cite 3 different explanation methods: GNNExpl, Zorro, PGM. The other explanation methods are not GNN specific. Yet there exists a large landscape of explanation methods that are particularly desiged for GNNs, to name a few:

SubgraphX: On Explainability of Graph Neural Networks via Subgraph Exploration, Hao Yuan et al. 2021.

GNN-LRP: Higher-Order Explanations of Graph Neural Networks via Relevant Walks, Schnake et al. 2021.

GraphMask: Interpreting Graph Neural Networks for NLP With Differentiable Edge Masking, Michael Schlichtkrull et al. 2021.

PGM-Explainer: Probabilistic Graphical Model Explanations for Graph Neural Networks, Minh et al., 2020.


- The authors restrict themself to GNN explanations methods that consider nodes or edges. There exists other approaches that consider walks (GNN-LRP) subgraphss (SubgraphX) or sequences of subgraphs (GraphMask) which are neither cited nor discussed. If the authors claim to develop metrics for general GNN explanation methods this has to be mentioned. Particularly because some evaluation metrics would be not directly applicable. For example the subgraph search G' in faithfulness becomes much more complex when not considering only edges or nodes.

- It remains unclear how the metrics are restricted to GNN explanations. Isn't this adaptable to any neural network, or even any ML model? Please discuss this.

- The actual contribution seems quite mild to me. Particularly because parts of faithfulness and sparsity has already been discussed in previous work (e.g. in Pope et al.). So in this view only correctness and plausibility is new to me. Please be more clear what is new and maybe also provide experiments which show how your metrics differ to those that already exist.

---

> ### Author Response · Authors · 2022-12-20
> **more experiment added, missed citation added and further clarification (1/2)**
>
> Thank you for your constructive comments. In the following we provide a few clarifications and details about changes made in the paper. We are enthusiastically looking forward to any further discussion.
> > The references are not complete. The paper is about evaluating explanation methods for GNNs, yet only very few GNN explanation methods are provided. In my opinion the authors only use and cite 3 different explanation methods: GNNExpl, Zorro, PGM. The other explanation methods are not GNN specific. Yet there exists a large landscape of explanation methods that are particularly desiged for GNNs.
>
> Thanks for pointing out the missed references. Unfortunately, the claim made by the reviewer is factually incorrect. We would like to point to section 2.1, 2nd paragraph of 1st version where we  have already cited SubgraphX, GraphMask and PGM-Explainer. Further, we would like to highlight that we used PGM-Explainer as a baseline in our experiments. The only missed citation provided in the list is GNN-LRP which we have cited in our revised version.
>
> >The authors restrict themself to GNN explanations methods that consider nodes or edges. There exists other approaches that consider walks (GNN-LRP) subgraphss (SubgraphX) or sequences of subgraphs (GraphMask) which are neither cited nor discussed. If the authors claim to develop metrics for general GNN explanation methods this has to be mentioned. Particularly because some evaluation metrics would be not directly applicable. For example the subgraph search G' in faithfulness becomes much more complex when not considering only edges or nodes.
>
>  Thanks for your feedback. As we mentioned above, we have already cited SubgraphX, GraphMask and PGM- Explainer in section 2.1, 2nd paragraph of 1st version.
>
> In this benchmark we do not add these methods. The reason why we did not add them as a baseline is because of applicability in our current setup as accepted by the reviewer. We have cited GNN-LRP in our revised version.
>
> Further, in our revised version, we use SubgraphX on synthetic dataset, the results are available in Table 8.
>
> >It remains unclear how the metrics are restricted to GNN explanations. Isn't this adaptable to any neural network, or even any ML model? Please discuss this.
>
> Standard Machine Learning (independent and identically distributed) tasks are specific cases of node classification where there is dependency between input samples. Hence these metrics are applicable to all general ML tasks. Further our metrics are applicable whether there is a structure or not. For example, similar metrics like  sufficiency and comprehensiveness are used in text domains (Eraser benchmark). Further the metrics like RDT-Fidelity and Sparity are applicable on any NN which works on datasets with input features. Only the RDT-Fidelity is not directly applicable because it operates over a computation graph, However If there is a notion of instance neighborhood, RDT-Fidelity can possibly be extended.  Also, [1] has used the decoys in the text domain.
>
> [1] “Towards Benchmarking the Utility of Explanations for Model Debugging”, Maximilian Idahl et al. 2021

---

> ### Author Response · Authors · 2022-12-20
> **novelty explained (2/2)**
>
> > The actual contribution seems quite mild to me. Particularly because parts of faithfulness and sparsity has already been discussed in previous work (e.g. in Pope et al.). So in this view only correctness and plausibility is new to me. Please be more clear what is new and maybe also provide experiments which show how your metrics differ to those that already exist.
>
> In choosing relevant metrics for the benchmark, we firstly started with the important metrics already proposed in the literature and categorized them based on their function. In this sense, we re-used existing metrics. However, the plausibility metric is our contribution. Also novel are graph classification datasets constructed from a text classification task, i.e., the movie reviews (MR) dataset.We note that converting them into graphs and considering it as a graph classification task for evaluating GNN explanations is indeed new. In addition, for measuring correctness, we propose the experimental protocols of adding decoys are also arguably novel and our contributions. Further we evaluate the explainers on (majority) real world datasets under one umbrella.
>
> **Why are we different from the metrics that already exist?**
>
> Pope et al.  proposed to compute faithfulness as the difference of accuracy (or predicted probability) between the original predictions and the new predictions after masking out the input features found by the explanation. This was called *Fidelity* in their work.
>
> Also, As the features cannot be removed in entirety to measure their impact, Funke et al  proposed  RDT-fidelity based on rate distortion theory defined as the expected predictive score of an explanation over all possible configurations of the non-explanation features.
>
> There are 3 ways where RDT-fidelity is different from fidelity from Pope et al.:
>
> 1. The major drawback of this metric is that, it is not guaranteed that new features are coming from the same distribution as of the original input features. Hooker et al. proposed the re-training of the model when input features are masked out.
>
> 2. The RDT-Fidelity is applicable locally in which we generate local datasets by applying multiple perturbations, which measures the robustness of the explanation.
>
> 3. It is also theoretically grounded by Rate Distortion theory.
>
>
> Pope et al. measure the sparsity of an explanation as the fraction of features selected by the explainer. Noting that this definition is not directly applicable for softmask approaches, Funke et al. proposes to quantify sparsity as entropy over the normalized distribution of explanation masks. We use the entropy-based sparsity metric as it can be applied both for hard and soft masking approaches.
>
>
> [2] Sara Hooker, Dumitru Erhan, Pieter-Jan Kindermans, and Been Kim. A benchmark for interpretability methods in deep neural networks, 2019.
>
> >Minor change
>
> Thanks for pointing out the error. We have modified the text accordingly.

---

### Review · Reviewer_XNZQ · 2022-12-07

**Summary Of Contributions:**

The authors propose a benchmark for GNN explanations. The study a large number of popular explainers, different models, and different datasets. They propose 4 different metrics (faithfulness, correctness, sparsity, and plausibility) that aim to capture various important aspects of what makes a good explanation.

**Audience:**

Yes

**Broader Impact Concerns:**

None.

**Claims And Evidence:**

No

**Requested Changes:**

- Evaluate node and edge level importance as given by the explainers without transformations. This means that the metrics would need to be adapted.
- Pareto plots that show the trade-offs between different metrics and from where it easier to read off whether a certain explainer dominates others.

**Strengths And Weaknesses:**

Strengths:
- The problem address in the paper is important since there is a pressing need for a unified and fair comparison of GNN explainer methods.
- The benchmark shift the focus away from toy datasets with planted explanations (which is the default evaluation).

Weaknesses:
- The authors transform edge masks into node masks by equally distributing the edge importance score. This is a major drawback. First, there is a bias towards high degree nodes. A high degree node with many unimportant edges may even end up being more important than a low degree node with a few important edges. Second, this implicitly assumes that the importance scores are additive. Finally, this is not evaluating the explanation methods on their own terms which might have been defined with edge importance in mind, making in unfair contrary to the motivation. More importantly, it is not necessary since some metrics can handle edge importance.
- It is not clear whether sparsity is a useful metric. The size of the true explanation -- the true subgraph that the model uses for prediction -- may vary for different graphs, and moreover may vary with the size of the graph (and the size of the k-hop neighborhood for node-level tasks). As an extreme example, we can create a model that randomly selects a subgraph of a varying size, throws away the rest of the graph and then makes a prediction. The true explanation should recover this subgraph regardless of the sparsity.
- In general none of the metrics are very convincing. Sparsity, plausibility, sufficiency and comprehensiveness have different issues as explained above and in the detailed comments below. Correctness, applies only to node-level homophobic tasks. RDT-Fidelity seems somewhat useful, but one can argue it tells us more about model stability than about the explanation. Moreover, it only applies to the node features. Extensions with a perturbation to edge features would be interesting.

Further detailed comments:
- The framing in the abstract and introduction of the paper is misleading. Based on wording and Figure 1, it appears as if though all metrics are applicable to all datasets and all tasks, but this is not the case.
    - For example, plausibility is only applicable where human rationale is available. The authors reuse the movie reviews dataset and construct an ad-hoc graph based on the text (all words within a sliding window of three are connected via edge). While this might be a good toy experiment it's hard to draw useful conclusions that generalize since e.g. the distribution of graphs is completely different from other "naturally" occurring graphs such as molecules.
    - The correctness metric only applies to node-level tasks. Moreover, it's only suitable for homophilic graphs (and models).
- It is not clear to me that the plausibility metric conceptually fits with the rest of the metrics or that is belongs in the proposed benchmark. As the authors state, plausibility measures how close is the decision-making process of the trained model (as revealed by explanations) to human rationales. However, this is a statement about the model and not the explainer. In other words, the explainer may be perfectly accurate, but this score can still be low because the way the model makes decisions is different. This is not to say that this metric is useless or that it has to be omitted, but at the very least this distinction compared to the other metrics needs to be discussed in more detail.
- As currently defined sufficiency promotes high confidence predictions. As the authors state a negative score signifies a higher $f(G')_j$. However, this does not seem to be desirable in all cases. For example, let's say that the graph is on the decision boundary between two classes (and that this is correct due to aleatoric uncertainty, i.e. the true $p(y|G)$ is uncertain). Then, we want the predicted distribution on the subgraph to also be similar. One measure for this would be the KL divergence between the predicted distributions over classes for $G$ and $G'$. Similar argument applies to comprehensiveness.
    - Relatedly, you might be able to manipulate these metrics by simply applying temperature scaling on the softmax, although this is property of the model and not the explainer.
- In Eqs. 2 and 3 the author say that they use the global empirical distribution of the features as the noise distribution. First, it is not clear how exactly this is computed and more details needed to be provided. Second, it is not well justified why this is a good choice and what is its implicit bias.
- Since some explainers produce hard masks, it is not clear why aggregated sufficiency/comprehensiveness makes sense for them. Moreover, in the aggregation equal weight is given to all bins which is not necessarily justified.
- The authors do not discuss the case of negative importance.

Minor comments:
- It might be helpful to give a short introduction to the evaluated explainers (e.g. in the appendix).
- The authors claim that they develop a unified, modular and extendable benchmark, but it is not clear what the modules are and how easy it is to extend them.
- The authors state that "there is no clear winner in GNN explanation methods". One can argue that based on the results Zorro is one candidate since for the models where it is applicable it does consistently perform in the top.

---

> ### Author Response · Authors · 2022-12-20
> **more experiments added and sparsity explained (1/n)**
>
> Thank you for your constructive comments. In the following we provide a few clarifications and details about changes made in the paper. We are enthusiastically looking forward to any further discussion.
>
> >The authors transform edge masks into node masks by equally distributing the edge importance score. This is a major drawback. First, there is a bias towards high degree nodes. A high degree node with many unimportant edges may even end up being more important than a low degree node with a few important edges. Second, this implicitly assumes that the importance scores are additive. Finally, this is not evaluating the explanation methods on their own terms which might have been defined with edge importance in mind, making in unfair contrary to the motivation. More importantly, it is not necessary since some metrics can handle edge importance.
>
> Thanks for your review. We agree that we do not explicitly model edge importance, but rather do it implicitly through node importance. We now have extended our work to consider edge importance by accounting for edge masks.Specifically, We perform an experiment on molecule datasets (MUTAG, PROTEINS and ENZYMES) where we use edge masks to calculate the sufficiency and comprehensiveness. We report the results in Table 16.
>
> We respectfully disagree with the comment that we have a bias towards high degree nodes. This is because we use a mean instead of a sum for aggregation.
>
> Also note that The RDT-Fidelity metric can be naturally extended to edge features. But in this benchmark we are not taking edge features into consideration. This is primarily because graphs with edge features are very common.
>
> In Plausibility and correctness, we need to retrieve node masks since the ground truth/ human rationales are available in terms of node masks.
>
> >It is not clear whether sparsity is a useful metric. The size of the true explanation -- the true subgraph that the model uses for prediction -- may vary for different graphs, and moreover may vary with the size of the graph (and the size of the k-hop neighborhood for node-level tasks). As an extreme example, we can create a model that randomly selects a subgraph of a varying size, throws away the rest of the graph and then makes a prediction. The true explanation should recover this subgraph regardless of the sparsity.
>
> Thanks for your comment. It is true that the size of the explanation can vary with the size of  the computational graph. It is therefore necessary to evaluate the quality of explanation using the faithfulness and sparsity metric. If two explanations have the same faithfulness, a sparser explanation should be preferred.
>
> Put simply, sparsity of our explanation refers to the property of a model or dataset that has a large number of zero or near-zero values in its feature space. In the context of interpretability, sparsity can be beneficial because it can make it easier to understand which features or variables are most important or influential in the model's predictions. As discussed in previous work[1], and we quote— **”Often for structured data, sparsity is a useful measure of interpretability, since humans can handle at most 7±2 cognitive entities at once.”**
>
>  This is also important because the explainer can retrieve the full input as an explanation which will be highly faithful. In section 5.2, we explicitly mentioned that faithfulness and sparsity of explanation should be analyzed together. We also report Sparisity and Fidelity in Appendix (Table 19) on ogbn-arxiv dataset, where we observe that a random explanation is highly faithful but the explanation is dense (sparsity is higher).
>
> [1] “Stop Explaining Black Box Machine Learning Models for High Stakes Decisions and Use Interpretable Models Instead” by  Cynthia Rudin.

---

> ### Author Response · Authors · 2022-12-20
> **futrther explanations on metrics  (2/n)**
>
> >The framing in the abstract and introduction of the paper is misleading. Based on wording and Figure 1, it appears as if though all metrics are applicable to all datasets and all tasks, but this is not the case.
> >* For example, plausibility is only applicable where human rationale is available. The authors reuse the movie reviews dataset and construct an ad-hoc graph based on the text (all words within a sliding window of three are connected via edge). While this might be a good toy experiment it's hard to draw useful conclusions that generalize since e.g. the distribution of graphs is completely
> >* The correctness metric only applies to node-level tasks. Moreover, it's only suitable for homophilic graphs (and models).
>
> To avoid the confusion, we have added Table 1, which contains information on datasets and metrics that we use in our benchmark. Further, the metrics are themselves not limited. Instead the applicability is rather dependent on presence/absence of ground truth for real world dataset. This benchmark is designed towards addressing challenges on the ground truth 1) by using the human rationales in graphs and 2) by generating ground truth in terms of decoys.
> * We additionally evaluate plausibility on Synthetic dataset as well where the ground truth is treated as human rationales.
> * The correctness metric is applicable on graph level as well. We are happy to extend correctness at graph level. But due to time constraints, we can not run experiments during rebuttal.
>
> >It is not clear to me that the plausibility metric conceptually fits with the rest of the metrics or that is belongs in the proposed benchmark. As the authors state, plausibility measures how close is the decision-making process of the trained model (as revealed by explanations) to human rationales. However, this is a statement about the model and not the explainer. In other words, the explainer may be perfectly accurate, but this score can still be low because the way the model makes decisions is different. This is not to say that this metric is useless or that it has to be omitted, but at the very least this distinction compared to the other metrics needs to be discussed in more detail.
>
> Thanks for your comment. You are absolutely right, this is the property of a model rather than an explainer.
>
> Plausibility measures the joint utility of the explanation method and the trained GNN model with respect to human rationales.
>
> In particular, plausibility only checks the agreement with the human rationales under the assumption that the explanations are faithful to the model. To effectively evaluate the GNN model via plausibility we should first check the faithfulness of the explainer.
>
> >As currently defined sufficiency promotes high confidence predictions. As the authors state a negative score signifies a higher f(G’)j. However, this does not seem to be desirable in all cases. For example, let's say that the graph is on the decision boundary between two classes (and that this is correct due to aleatoric uncertainty, i.e. the true p(y|G) is uncertain). Then, we want the predicted distribution on the subgraph to also be similar. One measure for this would be the KL divergence between the predicted distributions over classes for G and G’. Similar argument applies to comprehensiveness.
>
> While calculating the sufficiency and comprehensiveness, we only consider the probability of the predicted class rather than the distribution over classes.
>
> >In Eqs. 2 and 3 the author say that they use the global empirical distribution of the features as the noise distribution. First, it is not clear how exactly this is computed and more details needed to be provided. Second, it is not well justified why this is a good choice and what is its implicit bias.
>
> **How is the global empirical distribution calculated?**
>
> We sample the values from the underline training data distribution.
>
> **Why is this a good choice?**
>
> The reason for using the global distribution of features as the noisy distribution ensures that only plausible feature values are used.
>
> Besides, this choice does not increase the bias towards specific values, which we would have by taking fixed values such as $0$ or averages.
>
> The purpose of adding noise is not to replace the unimportant features of input with $0$, rather its value should not matter. The replacement of unimportant features with $0$ may cause side effects like in some datasets, the value $0$ may represent some semantic meaning or biasness towards some pooling strategy, for example, min pool. Also, the noise from global features distribution makes sure that the perturbed data points are still in the same distribution as the original data [2].
>
> [2] Sara Hooker, Dumitru Erhan, Pieter-Jan Kindermans, and Been Kim. A benchmark for interpretability methods in deep neural networks, 2019.

---

> ### Author Response · Authors · 2022-12-20
> **further explanation on sufficiency and comprehensiveness and response to minor comments (n/n)**
>
> >Since some explainers produce hard masks, it is not clear why aggregated sufficiency/comprehensiveness makes sense for them. Moreover, in the aggregation equal weight is given to all bins which is not necessarily justified.
>
> We only consider explainers generating soft masks for sufficiency  and comprehensiveness.
>
> We follow the bins strategy from the Eraser benchmark[3]. Since the different datasets’ instances may vary in size (i.e, number of nodes). The aggregated sufficiency and comp are calculated to make sufficiency and comprehensiveness comparable over different size datasets.
>
> [3] ERASER : A Benchmark to Evaluate Rationalized NLP Models, Jay DeYoung et al., 2020
>
> ### Minor Comments
> >It might be helpful to give a short introduction to the evaluated explainers (e.g. in the appendix).
>
>  We would like to point to Section B of Appendix of our 1st version where we have already provided introduction of the explainers.
>
> >The authors claim that they develop a unified, modular and extendable benchmark, but it is not clear what the modules are and how easy it is to extend them.
>
> Since this is an anonymous submission, we scale down our claims. We will be releasing our code/software publicly after acceptance where users can evaluate their explanation methods and datasets. Also, we will be releasing tutorials and python notebooks for evaluating new explanation methods.
>
> >The authors state that "there is no clear winner in GNN explanation methods". One can argue that based on the results Zorro is one candidate since for the models where it is applicable it does consistently perform in the top.
>
>  We would to highlight that this is not true, GNNExp outperforms Zorro on Correctness. Also on Synthetic dataset, we observe that SubgraphX outperforms Zorro in terms of sparsity and correctness (Table 8).
>
>
> ### Requested Changes:
> >Evaluate node and edge level importance as given by the explainers without transformations. This means that the metrics would need to be adapted.
>
> Thanks for your suggestion. We added new experiments where edge masks are directly used to evaluate sufficiency and comprehensiveness (Table 16).
>
> >Pareto plots that show the trade-offs between different metrics and from where it is easier to read off whether a certain explainer dominates others.
>
> We have added the plot in revised version (Fig. 5).
>
> We are enthusiastically looking forward to any further discussion.

---

### Comment · Reviewer_J5LF · 2023-01-12
**I find the contribution too small**

Thanks for commenting on the previous review. I'm still not convinced whether the paper has a significantly new contribution. Good luck!

---

### Decision · Action_Editors · 2023-01-20

**Recommendation:** Reject

**Comment:**

The paper proposes a new benchmark framework, called BAGEL, for evaluating explanations.  BAGEL consists of evaluation criteria and datasets, some of which are novel.  In the experiment, popular explanation methods are evaluated in the framework.

The current manuscript simply introduces the set of measures and datasets in BAGEL, and applied them to some explanation methods.  What is missing is to support the usefulness of BAGEL, i.e., importance of the measures and datasets, and convince readers that BAGEL will help push the field of XAI for GNNs.  Although BAGEL includes a new evaluation measure, the authors failed to convince reviewers about its usefulness.

Some previous papers proposed sets of evaluation measures, and the authors selected a new particular sets.  What is missing in the current manuscript is the discussion on what were problems with the previous sets, and how the selected ones remedy them.  Also, the paper should justify their choice of measures by arguing their optimality, necessity, and sufficiency (not necessarily rigorously though).  The authors should discuss what property is overlooked if one of the BAGEL measures is missing, why adding any of the unselected existing measures is redundant, and what happens if you replace one BAGEL measure with a similar existing one, etc.  Without such arguments, readers wouldn't be convinced that BAGEL would significantly contribute to XAI.  Similar arguments should be given for the selected datasets.  How do the selected datasets compliment each other, why is adding another dataset redundant?

I'd expect that the authors would really show the benefit of BAGEL in a future version, so that they can convince readers about the usefulness of BAGEL.



**Audience:**

The reviewers were not convinced that the proposed framework is useful.  In this sense, the target audience wouldn't be interested in the current paper.


**Claims And Evidence:**

Abstract simply says what is written in the paper without claiming, e.g., usefulness, of the proposed framework.  In this sense, the authors' claim is accurate.